

# Upper lithospheric structure of northeastern Venezuela from joint inversion of surface wave dispersion and receiver functions

Roberto Cabieces[1], Antonio Villaseñor[2], Elizabeth Berg[3], Andrés Olivar-Castaño[4], Mariano S. Arnaiz-Rodríguez[5,8], Sergi Ventosa[2,6], Ana M. G. Ferreira[7]

[1] Geophysical Department, Spanish Navy Observatory, San Fernando, 11408, Spain

[2] Institute of Marine Sciences, Pg. Marítim de la Barceloneta, 37-49, E-08003 Barcelona, Spain.

[3] Sandia National Laboratories, Albuquerque, New Mexico 87185, United States of America.

[4] Institute of Geosciences, Potsdam University, Karl-Liebknecht-Str. 24-25. Potsdam, Germany

[5] Departamento de Geofísica, Facultad de Ingeniería, Universidad Central de Venezuela

[6] Geosciences Barcelona, Geo3Bcn CSIC, c/ Solé Sabarís sn, Barcelona, Spain.

[7] Department of Earth Science, University College London, Gower place, WC1H 6BT London, UK

[8] Department of Physics of the Earth and Astrophysics, Universidad Complutense de Madrid (UCM). Madrid 28040, Spain

*Correspondence to*: Roberto Cabieces (rcabdia@roa.es)

**Abstract.** We use 1.5 years of continuous recordings from an amphibious seismic network deployment in the region of northeast South America and southeast Caribbean to study the crustal and uppermost mantle structure through a joint inversion of surface wave dispersion curves determined from ambient seismic noise and receiver functions. The availability of both ocean bottom seismometers (OBSs) and land stations makes this experiment ideal to determine the best processing methods to extract reliable empirical Green's functions (EGFs) and construct a 3D shear velocity model. Results show EGFs with high signal-to-noise ratio for land-land, land-OBS and OBS-OBS paths from a variety of stacking methods. Using the EGF estimates, we measure phase and group velocity dispersion curves for Rayleigh and Love waves. We complement these observations with receiver functions, which allow us to perform an H-k analysis to obtain Moho depth estimates across the study area. The measured dispersion curves and receiver functions are used in a Bayesian joint inversion to retrieve a series of 1D shear-wave velocity models, which are then interpolated to build a 3D model of the region. Our results display clear contrasts in the oceanic region across the border of the strike-slip fault system San Sebastian - El Pilar as well as a high velocity region that corresponds well with the continental craton of southeastern Venezuela. We resolve known geological features in our new model, including the Espino Graben and the Guiana Shield provinces, and provide new information about their crustal structures. Furthermore, we image the difference in the crust beneath the Maturin and Guárico Sub-Basin.



## 1 Introduction

The southeastern Caribbean and northeastern Venezuelan area (i.e., from 69°W to 60°W and 6°N to 14°N), located on the border between the Caribbean and South American plates, is a structurally complex area (Fig. 1) with active seismicity across multiple fault systems, the transition from oceanic to continental crust, a variety of sedimentary basins, and a continental craton.

The current Caribbean Plate configuration results from a transgressive evolution that began in the Tertiary and continued into the Cenozoic (Pindell et al., 1988; Meschede and Frisch, 1998; Müller et al., 1999), with a southern border dominated by strike-slip fault systems (Russo and Speed, 1994; Sisson et al., 2005). This region is also located between two major subduction zones: the Lesser Antilles to the east and south, and the Middle America trench to the west. One of the most important features of Northeastern Venezuela is the Eastern Venezuela Basin (Rohr, 1991). This structure, formed by oblique compression during the Oligocene to Miocene, is one of the world's most important petroliferous basins. The Guiana Shield, a continental craton formed during the Proterozoic-Archean, lies to the southeast of the basin (Fig. 1) and is one of the largest, oldest and highly stable tectonic features in South America. The western part of the Guiana Shield (formed around 3-2.8 Ga, as part of the Guriense orogeny) consists of metasedimentary rocks, including granitic gneiss and granitic intrusions that have been metamorphosed to amphibolite and granulite facies (Sisson et al., 2005). The eastern section of the Guiana Shield (2 – 2.7 Ga age), is primarily composed of metasedimentary rocks and mafic to felsic volcanic material, locally intruded by gabbro and diabase (Arnaiz-Rodríguez and Orihuela, 2013).

In terms of geodynamics, the plate interaction between the Caribbean and South American plates is evolving at a rate of 1-2 cm/yr westward (Pérez et al., 2001; DeMets et al., 2010; Webber et al., 2015), subducting the South American lithosphere beneath the Lesser Antilles (Ave Lallemant, 1997; Maria I. Jácome et al., 2003; J. Pindell et al., 2005). The detected seismicity in this region (as shown in Fig. 1 using data from 2005 to 2020) is sparse west of 65°, however an important cluster stretches between the Peninsula of Paria and the island of Trinidad. The earthquakes in this cluster range from shallow to intermediate depths (~ 40 to 150 km), and their magnitudes vary from Mw 3 to 5, with a few relatively large events (Mw ≈ 6.5). The Paria cluster contains a gap in seismicity between 36-51 km depth that Clark et al. (2008) used to conclude that the subducting and buoyant pieces of the South American Plate occurs along a near-vertical tear and supports a "jelly sandwich" rheology.

In this work, we study the structurally complex NE Venezuela region through Ambient Noise Tomography (ANT), which is a well-known tool developed in the last few decades (e.g., Shapiro et al., 2005) that is capable of imaging the crust and upper mantle. The most important step in ANT is the extraction of high-quality surface-wave empirical Green's functions (EGFs) from cross-correlations of seismic ambient noise (e.g., Bensen et al., 2007; Schimmel et al., 2011; Ventosa et al., 2019). Theoretical studies have proven that the EGFs between two points can be estimated from the cross correlation of the diffuse wavefield (e.g. ambient noise, scattered coda) recorded at the two locations when noise sources are well-distributed (Harmon





et al., 2007; Campillo et al., 2011). However, only a few studies, either at regional or local scales, use EGFs retrieved from
Ocean Bottom Seismometers (OBSs) to obtain ANT from onshore to offshore regions (Tian et al., 2013; Lee et al., 2014;
Corela et al., 2017; Ryberg et al., 2017; Hable et al., 2019). The use of ANT methodologies with OBSs presents several major
difficulties in comparison with continental stations. Frequently, seismic energy recorded by the OBSs is contaminated by low-
frequency oceanic infragravity waves (Webb, 1998; Bell et al., 2015; Tian and Ritzwoller, 2017). Additionally, the OBS
instruments can be affected by local conditions, such as interactions of ocean currents with the sea floor that can cause tilting
noise (Webb and Crawford, 1999; Deen et al., 2017)
The study area has previously been imaged using passive seismic techniques such as finite-frequency P wave tomography
(Bezada et al., 2010), analysis of 20-100s period earthquake-derived Rayleigh waves (Miller et al., 2009) and investigation of
Rayleigh wave phase velocities extracted from seismic ambient noise (Arnaiz-Rodríguez et al., 2021). Previous studies have
also resolved Moho depths in this area through receiver function (RF) analysis (Niu et al., 2007). In this work, we combine
the seismic ambient noise tomography approach with the information provided by receiver functions. Our joint inversion of
these two datasets provides more strongly resolved models than finite-frequency P wave tomography for the crust and upper
mantle. Additionally, our joint inversion overcomes limitations of surface wave tomography alone, including non-uniqueness,
poorly constrained shallow structure and poor sensitivity to interfaces, such as the Moho. These advantages enable us to gain
new insights into the study region.

**FIGURE 1**

Specifically, we use a joint inversion of ANT and RFs using a non-linear inversion based on the Markov Chain Monte Carlo
(MCMC) methodology. Through cross-correlating 1.5 years of data, we extract 6 to 32 seconds period fundamental mode
Rayleigh and Love waves recorded on 38 broadband stations. The results are discussed with reference to previous studies (Niu
et al., 2007; Miller et al., 2009; Bezada et al., 2010; Arnaiz-Rodriguez et al., 2021), and how our findings enable us to
characterize known tectonic features as well as obtaining new information.

## 2 Data and Methods

Data for this experiment were collected with a sampling rate of 100 Hz on 27 land stations (STS-2 / Quanterra-330) and 11
broadband OBSs (Trillium-40 sec) from the Ocean Bottom Seismograph Instrument Pool (OBSIP). The minimum interstation
distance is 30 km, and the aperture of the complete seismic network is approximately 1000 km. Both vertical and horizontal
component recordings are used from the land stations, but only the vertical component is considered from the OBSs due to
contamination of the horizontal components with oceanic noise (Webb, 1998). For the computation of the receiver functions
(RFs), all events with $M_w > 5.5$, and epicentral distances ranging from 30º to 90º, from March 2004 to March 2005 (~ 100





events) were considered. Theoretical first arrival times were computed for each event and station using the TauP library (Crotwell et al., 1999), and the continuous waveforms were cut 10 s before and 80 s after the computed first arrival times.

In the following sections, we briefly describe the steps we followed to build reliable EGFs from ambient seismic noise analysis. We also discuss how we obtain the RFs and the Moho depth estimation. We also review how we retrieve Rayleigh wave dispersion measurements from the cross-correlation of the vertical components of land and OBS station pairs (ZZ) and Love waves from the cross-correlation of the transverse components (TT) recorded by land stations. We closely follow the approach of Bensen et al., (2007) with some modifications in the preprocessing to obtain clear, reliable results for the OBSs. Subsequently, we create a set of phase and group velocity maps for the study region, and jointly invert the information contained in these maps with receiver functions to obtain our 3D shear-wave velocity model.

## 2.1 Empirical Green Functions (EGFs)

In the first preprocessing step, daily seismograms are analyzed and removed if they are found to contain glitches or gaps in more than 5% of the daily record to prevent artifacts in later processing steps. Next, we down-sample the seismograms to 5 Hz for computational efficiency. Then we remove the mean and deconvolve the instrument response (converting the seismograms to velocity) and rotate the horizontal components between stations to obtain the transverse component (T), except for the OBSs. This allows us to obtain estimates of the EGFs for the transverse components directly, from which we extract dispersion measurements for the Love waves. We apply absolute mean time normalization to remove unwanted earthquake contamination (Bensen et al., 2007) for each station on the vertical and transverse components independently. To form the temporal normalization function we apply a bandpass filter of 0.01-0.05 Hz and apply a running absolute mean with a window of 128 s. We divide the data from each station and component by the temporal normalization function. Next, we perform spectral whitening to expand the frequency band and balance persistent sources of noise (Ritzwoller et al., 2001; Lin et al., 2008). We divide the spectrum of each station and component by its smoothed, 0.025 Hz (20 points halfwidth), running-mean spectrum.

Finally, we compute cross-correlations on 4-hour segments of the preprocessed seismograms (Z-Z) and (T-T) for every pair of stations. Although the linear stack is the common approach to compute the EGFs, we prefer to consider the time-scale phase-weighted stack (ts-PWS) (Ventosa et al., 2017, 2019) to enhance the quality and increase signal-to-noise-ratio (SNR) of the EGFs. The complex frame used in the wavelet approach of the ts-PWS is a computationally efficient method for large datasets in comparison with the conventional PWS.

Fig. 2 displays both kinds of stacks (linear and ts-PWS) of the vertical components for different paths after filtering the EGFs with a bandpass filter (6-10 s). It is clear from Fig. 2 that, in general, the ts-PWS yields higher signal-to-noise ratio results than the linear stack, especially for noisy stations, such as OBSs or stations in areas with thick sediments (Cabieces et al., 2020).

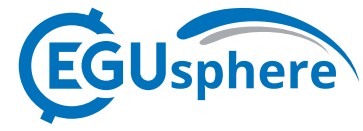

**FIGURE 2**
The EGFs estimates contain two signals, the causal (positive lag time) and the acausal (negative lag time), which are symmetric
in an ideal case. However, the spatial and temporal variation of the noise source distribution strongly affects the shape of the
EGFs in the time domain and may also induce perturbations of the spectral energy content. Fig. 3 displays the record section
of the Z-Z and T-T correlations with station CMPC, located in the Guiana shield. We see strong differences in amplitude
between the causal and the acausal portions for both components, but broadly these are more pronounced in the T-T
component. To reduce the influence of seasonal variations (Tanimoto et al., 2006; Schimmel et al., 2011) and potential impact
of oceanic waves dominating the noise field (Hillers et al., 2013), we average the causal and the acausal part of the EGFs
(shown in Fig. 3 for station CMPC, located in the Guiana Shield).

**FIGURE 3**

**2.2 Dispersion measurements of phase and group velocity**
We obtain fundamental-mode phase and group velocity estimates for both Rayleigh and Love waves in a period band ranging
from 6 to 32 s. We manually pick these dispersion curves using the Computer Programs in Seismology (CPS, Herrmann, 2013)
software. In order to use only the most reliable EGFs, we impose several criteria, including a minimum signal-to-noise ratio
(SNR > 40) in the period band T~0.5-100 s to reject unstable EGFs, and a minimum interstation distance of three wavelengths
(Boschi et al., 2013; Luo et al., 2015) to satisfy the far-field approximation.
Figure 4 summarizes the dispersion curves calculated from the vertical (Fig. 4a) and from the horizontal (Fig. 4b) components
for land-land, ocean-ocean, and land-ocean cross-correlations. This includes dispersion curves from cross-correlation of ARPC
to MAPC, two land stations that cross from the Guiana Shield toward the Maturin Basin and labelled in Fig. 1. The ARPC-
MAPC dispersion curve shows the typical behavior of a continental region, with low velocities slowly increasing in the
intermediate period range (T~10-30 s). Fig. 4 also shows the dispersion curve measured for the CUBA-MHTO pair, two OBS
stations, in which the most interesting feature is the higher velocities in the same period range, highlighting the crustal thinning
occurring towards the north of the study area and sensitivity of these measurements to the transition from the crust to the upper
mantle. Finally, results for the LAPC-PINA station pair in Fig. 4 illustrate the results obtained from the cross-correlation of a
land station and a OBS station, respectively. The LAPC-PINA station pair dispersion curve also shows a steep gradient curve
due to the path through the oceanic region, but its velocity is lowest among the other dispersion curves shown in this example.

**FIGURE 4**





The measurements of phase and group velocity from all station pairs in different periods can be used to form histograms that
help to visualize outliers (Fig. 4 and Fig. S1 in the Supplementary Material). These histograms show that our measurements
of group and phase velocity are most stable between periods of 6 and 32 seconds. The group and phase velocity values with
periods outside of a maximum cut-off of three standard deviations were discarded to remove erroneous measurements that
could negatively impact the tomographic inversion. For this same reason, those periods with less than 100 valid measurements
are also automatically rejected.

### 2.3 Station orientation and Moho Depth estimation

Accurate orientation of the horizontal components of the seismic stations is critical, both to obtain reliable receiver functions
and Love wave velocity estimates, especially when using OBSs. The SNR of the RFs calculated in a misoriented station is
generally diminished, and the peaks associated with the Moho depth can be shifted. We have oriented the OBSs used in this
work following the methodology of Doran and Laske (2017). Their method takes advantage of Rayleigh wave ellipticity to
orient the horizontal components. The goal is to find the maximum correlation between the Hilbert transform of the vertical
component and the horizontal component in a time window that contains a Rayleigh wave generated by a shallow teleseism.
Correct timing for the windows is determined using global dispersion maps of Rayleigh waves (Ma and Masters, 2014; Ma et
al., 2014).
To perform this analysis we use all available earthquakes with magnitudes raging from 5.5 to 8.0, epicentral distances ranging
from 5º to 165º and a maximum depth of 150 km (~ 100 events). Orientation results for the OBS stations and their associated
uncertainties are shown in the Table S1of the Supplementary Material.
One of the most important constraints required to obtain geologically reasonable solutions from the inversion of dispersion
curves of surface waves is a reliable Moho depth estimation, as the surface waves in the period range considered in this study
only weakly constrain the deeper crustal structure. We obtain Moho depth estimates across the study area through an H-k
analysis of the receiver functions (Zhu & Kanamori, 2000) computed for both the land stations and the OBSs, using the
Integrated Seismic Program (ISP) (Cabieces et al., 2022) as shown in section 3.2. First, we rotate the waveforms from the ZNE
to the LQT reference system. Since the back azimuth of the P-to-S converted arrivals might deviate slightly from that of the
great circle linking the event epicenter with the receiver, we investigate a suite of rotation angles following Wilde-Piórko et
al. (2017). We then compute receiver functions through the water-level deconvolution technique (Langston, 1979), which
suppresses instabilities in the spectral division of the L and Q components via the water-level filter. We also apply a Gaussian
filter in the deconvolution process to ensure that the resulting receiver functions do not suffer from misleading high-frequency
content (Langston, 1979). Depending on the SNR of the resulting receiver functions, the values of the water-level parameter
and Gaussian filter width used in this study range from 0.01 to 0.0001 and 2 to 4, respectively.



Finally, to incorporate the receiver functions into the joint inversion with the phase and group velocity of surface waves, we
determine the isotropic component of the receiver functions via harmonic decomposition (e.g., Shen et al., 2012). The isotropic
component represents the velocity contrasts directly below the recording station and is, in theory, reducing the effects of any
lateral heterogeneities (Bianchi et al., 2010). Moreover, the harmonic decomposition technique allows us to estimate an
uncertainty range for the receiver functions to be used in the joint inversion (see section 3.2).

## 2.4 Phase and Group velocity maps

We use the fundamental-mode Rayleigh and Love-wave group and phase velocity measurements to produce a set of isotropic
group and phase velocity maps following the inversion procedure described by Barmin et al. (2001) and implemented by
Olivar-Castaño et al., (2020). In this approach, we discretize surface wave velocities across the study area (phase or group)
along a regular grid (10x10 km in this study) and linearize the traveltime inversion by assuming that surface waves travel along
the great circle paths between respective station pairs. We minimize the misfit to the observed traveltimes to resolve the
velocity maps. We regularize the problem through multiple constraints, including a smoothing condition (controlled by a
parameter $\alpha$) and a penalization for deviations from the average velocity in areas of poor data coverage (controlled by a
parameter $\beta$). One of the advantages of this method is the ability to estimate data covariance and spatial resolution maps.
The traveltime inversion procedure is performed twice in order to remove outliers (Barmin et al., 2001). First, the phase
velocity maps are oversmoothed by setting a very high value for $\alpha$. Second, the observed traveltimes are compared with those
computed using the oversmoothed maps. The measurements corresponding to residuals greater than three standard deviations
are discarded (Moschetti et al., 2007).
We empirically derive the optimal inversion parameters to minimize model residuals and prevent sharp artifacts (Barmin et
al., 2001). Additionally, the final choice of regularization parameters ($\alpha$ = 600, $\sigma$ = 250, $\beta$=10) reproduce several smooth
features that are consistent with known geological features of the study area, as shown in maps of Rayleigh (Figure 5) and
Love (Figure 6) phase and group velocity results.
The ray-path coverage is shown in the Supplementary Material for both Rayleigh and Love waves. We highlight the ray-path
coverage and velocity maps for the periods of 12, 22 and 32 seconds for the Rayleigh wave group and phase (Figs. S2 and S3,
respectively) measurements. We also include the Love wave group and phase (Figs S4 and S5) ray-path coverage and velocity
maps for 12 and 20 s periods. We use the ray-path coverage and spatial resolution maps (Figs. S8 and S9 in the Supplementary
Material) as a proxy to determine where we expect reliable group and phase velocity results. In order to further evaluate the
limitations of our group and phase velocity maps, we performed a checkerboard test (Fig. S6 and S7 in the Supplementary
Material). The size of the velocity anomalies that we try to recover in the checkerboard tests is period-dependent (equal to one
wavelength). The results of these tests indicate stronger constraint of velocity anomalies in the south (Guarico and Maturin



basins) than in the region covered by OBSs, and overall we see higher spatial resolution for Rayleigh waves than for Love
waves.
**2.5 Joint Inversion for Shear Wave Velocity**
We perform a 1-D inversion at each point within the grid with a resolution of 0.5º x 0.5º (from 68.5ºW to 60.5ºW and from
5.5ºN to 14.5ºN) using the corresponding phase and group velocity dispersion curves extracted from the maps discussed in
section 2.4. Next, we jointly invert the dispersion curves and the receiver functions, in the closest grid points to a station
(distance < 0.5º) using the Markov Chain Monte Carlo (MCMC) Bayesian inversion tool BayHunter (Dreiling et al., 2020).
This algorithm is inspired by the methodology of Bodin et al. (2012), in which the Bayesian formulation is applied to foster a
posterior probability distribution, where each model parameter can be described with a full probability density function.
Bayesian inversions have an important advantage over deterministic methods due to their exploration across a wide range of
velocity models, making the choice of initial models less critical. Additionally, Bayesian inversions enable calculation of final
model uncertainty from the posterior distribution and determination of data sensitivity to the model space (Shen et al., 2012;
Berg et al., 2020).  In addition to obtaining the velocity-depth structure, the BayHunter algorithm resolves the probability
distribution of Vp/Vs ratio, the optimal number of layers of the model and the noise parameters (i.e., data noise correlation
and amplitude) within a user-defined range.
In terms of inversion parametrization, we searched over a range of shear wave velocities (Vs) from 2 to 6 km/s (based on
existing geological information), Vp/Vs from 1.6 to 1.9, a maximum of 20 layers and a decreased and increased tolerance
limits of Vs with respect to depth of 10% and 75%, respectively.  The uncertainties amplitudes $\sigma$ Surface Wave Dispersion
Curves (SWDC) and $\sigma$ RF were set from 0 to 0.1 km/s and the correlation parameter for SWDC 0.2 and for RFs 0.96.
While our MCMC joint inversion includes both surface wave and receiver function data with the objective of a best fit Vs
model, we also include a prior constraint on the Moho depth. The Moho depth is obtained in the H-k analysis of the receiver
functions. At each inversion point, we ran a total of 40 Markov chains with a final distribution of 100,000 iterations per chain,
keeping all models within an accepting rate of 40% to form the posterior probability distribution. We tried those range of
values for the set of inversion parameters , in line with accepting rates of previous studies (e.g., Bodin et al., 2012; Dreiling et
at., 2020) and we found that they led to stable results and reasonable computational efficiency.
**3. Results**
**3.1 Phase and Group Velocity Results**
The fundamental-mode phase and group velocity maps of Rayleigh (Fig. 5) and Love (Fig. 6) waves show patterns related to
the main geologic structures, including the Maturin Basin (Moho depth $\approx$ 45 km) and Guiana Shield (Moho depth $\approx$ 35 km).
Both sets of results in Figures 5 and 6 are displayed as % perturbation of the mean velocity for three representative wave





periods, with the corresponding spatial resolution maps in the Supplementary Material (Figs. S8 and S9). Relatively low phase
and group velocities are recovered for the Guarico and Maturin sedimentary sub-basins. These low-velocity anomalies are
stronger towards the southern parts of these basins, suggesting increased sediment thickness typical of foreland basins.
However, these low velocities are not observed south of the basins in the transition to the Guiana Shield (Maria Ines Jacome
et al., 2003). The contrast between the lower relative velocities of the Maturin basin and the higher velocities of the Guiana
Shield are especially evident in the phase and group velocity maps of Love waves (Figs. 6a-f), which are more sensitive to
near-surface structure than their Rayleigh-wave counterparts (Lin et al., 2008).
The velocity contrast between the oceanic crust and the sedimentary basins can also be seen in the Rayleigh wave group and
phase velocity maps, most prominent at longer periods (Fig. 5 b-c, e-f). However, as the depth sensitivity of the surface waves
increases with increasing period, this contrast becomes more apparent with increasing periods (e.g. Fig. 5f). We also note that
the San Sebastian - El Pilar fault system appears as a sharp boundary, separating regions of different velocities in the NE along
the Peninsula of Paria - Trinidad and Tobago (see Figs. 5e and 5f).
**3.2 Receiver Functions and Moho Depth Estimation**
To study the Moho depth and its gradient, we split the analysis into three separate regions, the south of the Venezuela Basin,
Guarico - Maturin Basin and the Guiana shield. Fig. 7 shows the map of the Moho and the Vp / Vs ratio estimated applying
the methodology of section 2.3 and Fig. 8 shows three examples of RF stacks and H-k maps for stations placed in the Venezuela
Basin, Guarico - Maturin Basin and the Guiana shield (stations MHTO, PRPC and CAPC, respectively).

71                                                           **FIGURE 7**

Overall, the receiver functions obtained from the OBSs stations deployed in the Venezuela Basin and the land station in
Margarita Island are of poor quality, so we ensure to use only the most robust RFs in the inversion. Despite limited data, we
estimate Moho depths for this area that are consistent with expected depths, which range from 20 to 32 km (Niu et al., 2007),
and we resolve a shallower Moho (< 20 km) beneath the northwestern coast. We also find that the Moho depth around
Margarita Island increases to approximately 40 km. In general, we find that the north Caribbean crust has a constant thickness
from the north Caribbean plate to the coast. Additionally, our results reveal thin crust southwest of the Venezuela Basin
(stations SHRB-PNCH-CUBA-PINA) that slightly differs from the work of Nui et al., (2007) but agrees with the expected
thickness of the Venezuelan Abyssal Plain (Officer et al., 1959; Edgar et al., 1971; Romito and Mann, 2020).
Our results show Moho depths across the Guarico - Maturin Basin (Figure 8, stations STPC and PRPC) ranging from 45 km
to 50 km, which agrees with the previous work of Nui et al., (2007). However, we observe an anomalously thickened crust of
up to 50 km beneath the southeastern Orinoco River region (stations PRPC and PAPC). The signature of this thickened crust
can be correlated with a similar NW to SE elongated feature present in the group and phase velocity maps.



Lastly, we obtain receiver functions with the highest SNR using data from stations in the craton (Fig. 8, station CAPC). Using
these high-SNR RFs, we estimate a crustal thickness of approximately 40 km, which is lower than the values recovered for
the Guarico and Maturin basins to the north.
89                                                **FIGURE 8**

**3.2 Shear velocity model**
We build a 3D shear wave velocity model on a 0.5º x 0.5º grid with depth increments of 0.5 km from the 1D models obtained
from the joint inversion of the phase and group velocities and the receiver functions. We focus on the region between 68ºW-
61ºW and 6ºN-14.0ºN, corresponding to the area in which the highest density of surface wave velocity and Moho depth
estimates are available and well-constrained. For each grid point, we invert the phase and group dispersion curves of Rayleigh
and Love waves from 10 to 40 s periods in addition to the receiver function corresponding to the nearest station. We allow a
maximum distance to a grid point of 0.5º for the RFs, but we do not apply additional weights to the receiver function data
according to the distance from the grid point to the nearest station.
Fig. 9 shows an example from a location in the Guiana Shield (65.0° W, -7.0 ° N) of the results obtained from the 1-D joint
inversion of RFs and Rayleigh and Love wave phase and group velocity data. Specifically, the left panels (Fig. 9a) contain the
assembled shear-velocity model obtained after the search of 100,000 possible models in each of the 40 Markov chains. Figure
9b-e displays the predicted data from the inversion of phase and group velocity data and uncertainties thereof, while Fig. 9f
shows the fit to the isotropic receiver function obtained from station MAPC.
We observe some discrepancy between the different datasets, as evidenced by the difficulty to perfectly fit all the data.
However, despite some discrepancy between the fit to different datasets, which causes minor perturbations in the 1D profile,
we observe an overall good fit to the dispersion curves and to the RF for each gridpoint.
07                                                **FIGURE 9**

Fig. 10 shows un-smoothed horizontal slices of our shear-wave velocity model at different depths ranging from 7.5 to 60 km.
We also display uncertainty maps of our joint inversion, corresponding to one standard deviation of the mean MCMC joint
inversion result, extrapolated for each 1D grid point model (Fig. S11 in the Supplementary Material). Furthermore, we include
maps comparing the horizontal slices of the 3D model, made of MCMC joint inversion results across all 1D gridpoints with
robust data, with the reference global model ak-135-f (Kennet et al., 1995) (Fig. S12 in the Supplementary Material).
To further explore our results with depth we include a set of south-to-north and west-to-north oriented depth cross-sections of
the model at different longitudes between 61° W - 68° W and 6° N - 14° N in Figure 11. In the shallower layers of the shear-
velocity model (≈ 15 km), we observe sharp velocity variations and a clear contrast between the Venezuela Basin with low



absolute shear velocities and the Guiana Shield with high shear velocities, which is expected for a continental craton. In the
Guiana Shield area (Fig. 10 a), we observe near-surface faster shear velocities. On the other hand, one of the most remarkable
features we see in the cross-sections (66º W – 62ºW) is the presence of a slightly low velocity anomaly from 9º N  to 14º N
between 8 km and 16 km depth, being the strongest in the 64ºW cross-section (Fig. 11 B-B').

22                                                **FIGURE 10**

We observe a high shear velocity structure through intermediate depth slices of our modeled results (15 to 45 km depth) below
the limited area surrounded by the Tortuga Island - Barcelona Bay and Margarita Island (Fig. 10 b-f). This high velocity
structure persist up to approximately 45 km depth and is also prominent in the cross sections (Fig. 11 d-f) from 64ºW to 66ºW.
The deepest parts of our model do not show any remarkable features except for a persistent high-velocity anomaly in the
Barcelona Bay that extends towards the SE beneath the Maturin Basin.  In general, we find that shear-wave velocities increase
with increasing depth for all regions, with no remarkable velocity inversions. Model cross-sections depict relatively constant
Moho depths between 8º N to 12º N across the study area, with an interesting dipping structure at around 10º N, suggesting a
thickened crust towards the south. Additionally, we see low velocities corresponding to the thick Maturin Basin at 10ºN.
Finally, similar to the results of Arnaiz-Rodríguez et al., 2021, we also detect an anomalous low velocity in the lithospheric
mantle beneath the Peninsula of Paria that expands from 15 km to 22 km depth.

36                                                **FIGURE 11**

**4. Discussion**
Vertically and horizontally polarized dispersion measurements from ambient noise records and teleseismic receiver functions
enabled us to investigate the variations of crustal thickness, seismic velocities, and Poisson's ratio in northeastern Venezuela.
Robust velocity values and anomalies (Fig 10, Fig. 11 and Fig S12 in the Supplementary Material), as well as a detailed Moho
(Fig. 7a) and Vp/Vs maps (Fig. 7b) allow us to image the region in a clearer way than in previous studies (e.g. Niu et al., 2007;
Miller et al., 2009; Arnaiz-Rodríguez et al., 2021). For the first time, we image the upper lithosphere using a joint inversion
of dispersion measurements and RFs with a robust non-linear inversion. Our results are correlated to well-known geological
structures and tectonic provinces and shed new insight into the crustal structure of the region. We begin our discussion by
taking a look into the results derived from receiver function analysis and then we examine the shear-wave model product of
the inversion of the dispersion curves.



### 4.1 Moho Depth from Receiver functions

The estimated Moho depth in the study area ranges from 10 km to 51 km. Shallower Moho depths (<20 km) are clearly associated to the Caribbean basin at the North, while the transitional crust and inactive volcanic arcs found between the oceanic plateau and the continental shelf are characterized by a Moho depth ranging between 20 km and 30 km. Interestingly, the Serranía del Interior Mountain range appears in the map within this region in agreement with previous estimations (e.g. Niu et al., 2007) even though no stations were used inside the range in this study.

We find that the Moho depths on the Venezuelan continental shelf range from 30 to 51 km. We find the region with the greatest Moho depths in the southernmost Guárico Basin, with results similar to the values reported by Niu et al (2007). However, our results for the area with thickest crust are counter-intuitive with standard foreland basin configurations, where the depocenter of the basin is associated with the deepest flexure and the deepest Moho (e.g. Watts, 2001). In the Eastern Venezuela Basin, the deepest sediments (~10 km; Feo-Codecido et al., 1984; Di Croce et al., 1999; Clark et al., 2009; Bezada et al., 2010) are found in the Espino Graben and Maturin sub-basin located north and northeast of the region with greatest Moho depths. We find the trend and location of the thickest crust is relatively consistent with the location of the contact between the Precambrian Basement (i.e., the Guayana Shield extending beneath the sedimentary layers North of the Orinoco River) and the Cambrian basement found between the Espino Graben and the Shield (e.g., Feo-Codecido et al., 1984; Di Croce et al., 1999). We suggest this indicates the allochthonous palaeozoic terrains accreted to the North of the Guayana Shield either had a crustal thickness larger than the shield itself, or, alternatively, that the accretion process deformed the crust in such a way that now the deepest Moho is found on this contact. Finally at the South, the Guayana shield region is characterized by Moho depths ranging between 35 and 40 km. The largest values are consistently associated with the Imataca province (3–2.8 Ga, Guriense) while the Pastora province (2.7– 2 Ga, Pre-Transamazonic) corresponds to a region with the thinnest crust in the shield at an average crustal thickness of 36 km. These results agree with those reported by Niu et al. (2007).

### 4.2 Vp/Vs Map

Unlike previous attempts (e.g., Niu et al., 2007; Masy et al., 2015), we can consistently map the variations of the average crustal Vp/Vs in the region. In the shield we found an average value of ~1.72 Vp/Vs (roughly 0.26 in Poisson's ratio), consistent with the global average for continental crust (Christensen and Mooney, 1995). Similar to Niu et al (2007), we find this value to be lower than expected for ancient shields (e.g. Zandt and Ammon, 1995). Niu et al. (2007) attempted to explain this anomaly by suggesting that the crust in the region presents low quantities of granulite-facies mafic rocks. Unfortunately, mafic rocks are rather common in the Imataca (Dougan, 1977) and Pastora (Ostos et al., 2005) provinces. Rather, we propose that the presence of low Vp/Vs can be attributed to the relatively large amounts of quartz in the upper crust in this area, which is consistent with the commonly reported felsic volcanism across the region. Lower values of Vp/Vs (<1.65) are found in the Espino Graben and Paria Peninsula. In both regions mafic rocks have been reported as: (a) basaltic layers found inside the



Espino Graben (e.g. Feo-Codecido, 1984) and (b) the dacitic rocks (porfidic rhyolite; Alvarado, 2005) found in the *Serranía*
*del Interior*. Our results appear inconsistent with the lithology of the regions, suggesting the values are most-likely related to
the high density of faults in both areas.

### 4.3 S-Wave Velocity model.

The shear wave velocity structure of the Eastern Venezuelan crust and upper mantle behaves consistently with what is expected
for the region. Lower velocities are found at the top of the cross sections (Fig. 11), gradually increasing with depth, with
different trends associated with each of the four main terrains: (a) The Caribbean at the North, (b) The heavily faulted and
deformed strike-slip dominated plated boundary, (c) the foreland basin and (d) the Guayana shield at the South. Mean crustal
shear-wave velocities (Vs) found in the continental shelf are around 3.99 km/s with a standard deviation of 0.64 km/s, which
is higher than the global average (~ 3.71 km/s according to Laske et al., 2013). The average crustal values for the oceanic
region are approximately 3.57 km/s with a standard deviation of 0.56 km/s, also slightly higher than the world's crustal average
(~3.383 km/s; Laske et al., 2013).
At shallow depths (7.5 km - 15.0 km, Fig. 10a and Fig. 10b) our model shows higher Vs values in the Guayana shield (mean
of ~3.6 km/s) and lower values in the foreland basin (mean of ~3.1 km/s). We note that the higher Vs of the Guayana shield
extends north of the Orinoco River in agreement with results obtained from well perforations (e.g. Feo-Coecido et al., 1984).
Lower Vs values found north of latitude 9° correspond to the foredeeps and depocenters of the basins that reach more than 10
km depth (Feo-Codecido et al., 1984; Di Croce et al., 1999; Clark et al., 2009; Bezada et al., 2010). We also found a low
velocity anomaly beneath the Espino Graben (~8 km depth), most likely associated with the large number of faults and thick
sedimentary layers therein. A similar anomaly appears in the 10 km depth slice (7.5 km Fig. 10a and Fig. 10b), except for the
appearance of a previously unreported high velocity anomaly within the Graben, being this anomaly well resolved in our model
(see also Figs S8 and S9 in the Supplementary Material for detailed Checkboard tests). We suggest that this may be the seismic
expression of basaltic intrusions at upper crustal levels that mark the opening of the graben and the intrusion of mafic magmas
into the continental crust during the Jurassic (162 ±8 ma; Feo-Coecido et al., 1984).
At the bottom of the upper crust (~25 km according to Clark et al., 2009) around 22.5 km (Fig. 10c and Fig. 10d) the Guayana
shield appears rather homogenous with and average Vs of ~3.6 km/s. At this depth our model shows Vs variations that correlate
well with differences in age of the different geological provinces in the basement of the Eastern Venezuela. The Precambrian
terrains of the once called "the Piarra block," a large area of uplifted Precambrian basement that occupies the Maturin Basin
(Feo-Coecido et al., 1984), show higher Vs than the Cambrian allochthonous terrains beneath the Guarico Basin. Hence the
petrology of these terrains are likely dissimilar. At 30 km to 37.5 km depth (Fig 10.d and Fig. 10e) we see the most prominent
anomaly to be the low velocity channel that perfectly aligns with the major faults of the Espino Graben major faults. Congruent
with the finding of Arnaiz-Rodríguez et al. (2021), we find that this region of high faulting appears as a low S-wave velocity
region (e.g. Yudistira et al., 2017). At this depth we also find a dichotomy within the uppermost mantle of the Caribbean



basins, with relatively higher Vs values (~4.2 km/s) beneath the Venezuela basin, whereas the Grenada basin is associated
with lower Vs (~3.9 km/s). These variations reflect the different histories of both basins: while the Venezuela basin is a
thickened remnant of the original Jurassic Caribbean crust, the Grenada basin is a back-arc basin that opened due to slab-roll
back during the Paleocene. From our tomography model alone, it is not possible either to distinguish if the observed anomalies
are related to the chemical differences or if the mantle beneath the younger basin is hotter in comparison to that of the older
one. Both anomalies can be observed in the model by Arnaiz-Rodríguez et al. (2021).
The lower crust (from 30 km to 50 km depth) appears rather homogeneous with small localized heterogeneities (with
lengthscale of about 7.5 km Fig 10d, 10e, 10f and 10g). One feature that stands out is the "U" shaped anomaly that aligns with
the Urica strike-slip fault and the Pirital frontal thrust (south of the Serranía del Interior; e.g. Audemard, 2006). This anomaly
implies that the faults extend deep into the crust and represent a major crustal limit. Finally, the uppermost mantle beneath the
continental shelf shows a large high velocity anomaly (deviation of ~20%, see Fig. S12). Part of this anomaly again follows
the general trend of the Espino Graben, perhaps a testament of the anatexis produced by the lithospheric thinning driven by
the opening of the Espino Graben. The largest anomaly, found West of the Paria Península, largely disagrees with a low
velocity anomaly found by Miller et al (2009) in the same region, but agrees with another high velocity anomaly reported by
Arnaiz-Rodríguez et al (2021). This anomaly is not related to any known feature in the region and could represent a chemically
anomalous region in the upper mantle, so it is perhaps a remnant of the southernmost prolongation of the Antilles subduction
beneath South America (e.g. VanDecar et al., 2003; Bezada et al 2010).
We present a detailed picture of the crustal structure across four profiles (Fig 11) containing shear-wave velocity variations as
well as the Moho depth. The first profile (A-A'), along -66°, shows, for the first time, the crustal structure beneath the
Cuchivero Province, with an upper crustal layer of ~ 20 km and with a homogeneous lower crust. The lower mantle in this
section appears with a smooth velocity gradient increasing from 4.25 km/s (at the Moho) to 4.75 km/s at 60 km depth. North
of the Orinoco River, the crust thickens from 40 to 50 km, with the maximum depth coinciding with the position of the Altamira
Fault (the physical limit between the southern Precambric terranes and the northern Paleozoic ones) reported by Feo-Coecido
et al (1984). The crust in this area appears well differentiated into 4 layers: an upper crust of ~ 20 km, a thin middle crust
(thickness < 10 km), and two lower crustal layers with high velocity when compared to the global average for continents of ~
3.93 km/s (Laske et al., 2013). When compared to the simpler crust found in the south of our study region, we find increasing
evidence to the collision and thickening of the crust and suture of two blocks during the emplacement of the Paleozoic terranes
to the Guyana shield. Further to the North, Vs in the upper crust decreases near the *Cordillera de la Costa* range, where also
low Vp velocities were also reported via wide angle seismic profiling (Magnani et al., 2009). Most likely, this represents the
nappe systems formed from the Paleocene to the Eocene by the collision of the Caribbean arc with northern South America.
Here, the Moho gradually thins out up to the San Sebastian Fault System, which separates the extended transitional crust
(originally an arc) from the continental platform (Passalacqua et al., 1995). Finally, in the northernmost section of the profile,
we find the thick crust of the beneath the Bonaire Basin similar to the values reported by Bezada et al. (2010).



The second profile (B-B') shows a similar structure, but we highlight a few important differences. Firstly, we find possible
evidence of intrusions/chemical heterogeneities at the base of the crust of the Guyana shield, in agreement with geological
observations that report intrusions of different types of igneous rocks (from granitic to ultra-mafic) and greenstone belts rocks
(Sidder, 1990) that commonly occur in the Imataca Province. On this profile, the maximum Moho depth does not correspond
with any known structure, but could reflect the prolongation to the east of the feature reported in profile A-A'. North, towards
the *Serranía del Interior* and Margarita Island, the Moho is mostly constant within the contour of 4.25 km/s, but this changes
due to a lack of data further north, where the Moho beneath the transitional crust is better defined by the Vs model than by the
interpolation of the receiver functions data. Interestingly, this profile shows a high velocity anomaly and small shortening in
the continental crust beneath the *Serranía del Interior* and the Maturín basin that could be consistent with a NE projection of
the Espino Graben. Previously these features have only been ineffectively found through heavily processed aeromagnetic data
interpretation (e.g. Gonzalez et al., 2017), and in our model have been found thanks to the combined analysis of the moho
depth, RFs and dispersion measurements.
The Espino Graben structure is better appreciated in profiles C-C' (along the axis of the graben) and D-D' (perpendicular to
the structure). Low velocity anomalies are reported along the structure, as well as crustal thinning, a remnant of the opening
of the graben. Also, high velocity anomalies can be appreciated in the lower crust in profiles B-B', C-C' and D-D,' most likely
revealing the inclusion of mafic material during the extension process that created the Espino Graben. Most of the anomalies
seem to have some smearing to the North, but as the graben area is well covered (Figs S7 and S8 in. the Supplementary
material, which show resolution maps) this is unlikely to be produced by limitations in ray path coverage (see also Figs S2,
S3, S4 and S5 in the Supplementary Material, which show good resolution in this region). Therefore, we suggest that the
master fault of the graben deepens to the North, corroborating previous findings from Arnaiz-Rodríguez et al. (2021). As high
velocity anomalies consistently appear beneath the graben we cannot disprove the possibility of trapped material within the
lithospheric mantle, which is a hallmark of the process disproved above. Moreover, in profile D-D' we also see the crustal
structure beneath the Pastora province, that clearly shows a differentiation in upper, middle and lower crust.
Profile E-E' (in W-E direction) shows the crustal difference between the two sub-basins that compose the Eastern Venezuela
basin. Long have geologists tried to establish the difference between both, and in general the most important difference is that
the Guárico basin (to the West) sits on top of a Paleozoic age, while the Maturín one (to the East) formed over a Precambric
basin (the Piarra block, after Feo-Coecido, et al., 1984) where granite was found bellow the sediments at 4.3 km depth. Here,
we show that the upper-crust-lower crust discontinuity and the Moho appear to be rather flat (as would be expected for an
along-the-structure profile in a foreland basin). We find that the main difference is a large high velocity layer in the lower crust
beneath the Maturín basin. This kind of anomaly is similar to those found at the southern end of profiles A-A' and B-B',
reinforcing the long-lasting geological interpretation that the Precambian layers of the shield extent to the North beneath the
eastern section of the basin (e.g. Yoris and Ostos, 1997).





## 5. Conclusions

We present a new, 3D shear-wave velocity model of the SE Caribbean area built from a dense set of 1D models (spaced in a 0.5 ° x 0.5 ° grid). The 1D models were obtained from a nonlinear joint inversion of phase and group velocities of surface waves and receiver functions, using a Markov Chain Monte Carlo algorithm (Dreiling et al., 2020). Additionally, we provide a new Moho depth map obtained from a semblance-weighted H-k stacking analysis of receiver functions, which shows remarkable tectonic features. From the interpretation of these results, we outline the following conclusions:

A. The Moho interface in the region area ranges from 10 km to 51 km. Values <20 km are associated with the Caribbean crust, the transitional crust ranges between 20 km and 30 km and the Venezuelan continental shelf ranges from 30 to 51 km. A small shortening in the continental crust is reported inside the Espino Graben, testament to the opening of the Graben. A thickening of the crust, north of the Orinoco River, seems to be closely associated to the collision between Precambric and Paleozoic basement blocks. This indicates that the collision and subsequent suture thickened the crust and this structure is now buried beneath the sediment of the foreland basin.

B. The crust of the Guiana Shield is drastically heterogeneous. Variations are related to each of the major tectonic provinces therein. From West to East we find that: (a) the Cuchivero Province presents an upper crustal layer and a homogeneous lower crust, (b) the Imataca province shows similar upper crust to the Cuchivero Province but a heterogeneous lower one with anomalies possibly related to intrusions at the base of the crust, and (c) the Pastora province appears to show a differentiated section divided into upper, middle and lower crust.

C. The Espino Graben is associated with low values of Vs in the sedimentary layers ( < 3.0 km/s) and in the upper crust ( < 3.5 km/s), which are related to the large number of faults in the region. Furthermore, we find several high velocity anomalies between 10 and 15 km and between 20 and 25 km ( > 4.0 km/s). These are likely related to the basaltic rocks that intruded the crust during the opening of the Graben in the Late Jurassic.

D. We report high velocity bodies as well as a thinner crust beneath the Espino Graben, remnants of the extension that formed the structure. Furthermore, we found the first seismological evidence that the graben extends beneath the Serranía del interior Range.

E. The Vs in the upper mantle of the Caribbean Basins in the region is drastically different: ~4.2 km/s at the Venezuela basin, and ~3.9 km/s at the Grenada basin. This difference arises from the different age (old and cold lithospheric keel should have a larger Vs) and geological history.

*Data.* The facilities of IRIS Data Services, and specially the IRIS Data Management Center, were used for access to seismic waveforms and related metadata.



*Author Contribution.* Authors contributions to the different Sections and Subsections of this study, including data analysis and
interpretation, preparation of the figures and results, as well as writing, is as following:  Subsection 2.1 Empirical Green
Functions (EGFs): RC, AV and SV. Subsection 2.2 Dispersion measurements of phase and group velocity: RC and AV.
Subsection 2.3: AOC. Subsection 2.4: Phase and Group Velocity maps: RC, AV and AOC. Subsection 2.5 Joint Inversion for
Shear Wave Velocity: RC, AF and EB. Section 4. Discussion: RC and MA. AV, EB and AF devised and finalized the original
manuscript.
*Competing interests.* The authors declare that they have no conflict of interest.
*Acknowledgements.* Vs 3D-model, Empirical Green Functions, Receiver Functions Moho depth estimations and Dispersion
Measurements can be required to the corresponding author: rcabdia@roa.es. Figs 1, 7, 10 and 11 were generated with the
open-source mapping toolbox GMT (Wessel et al., 2013)



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



**Figures**


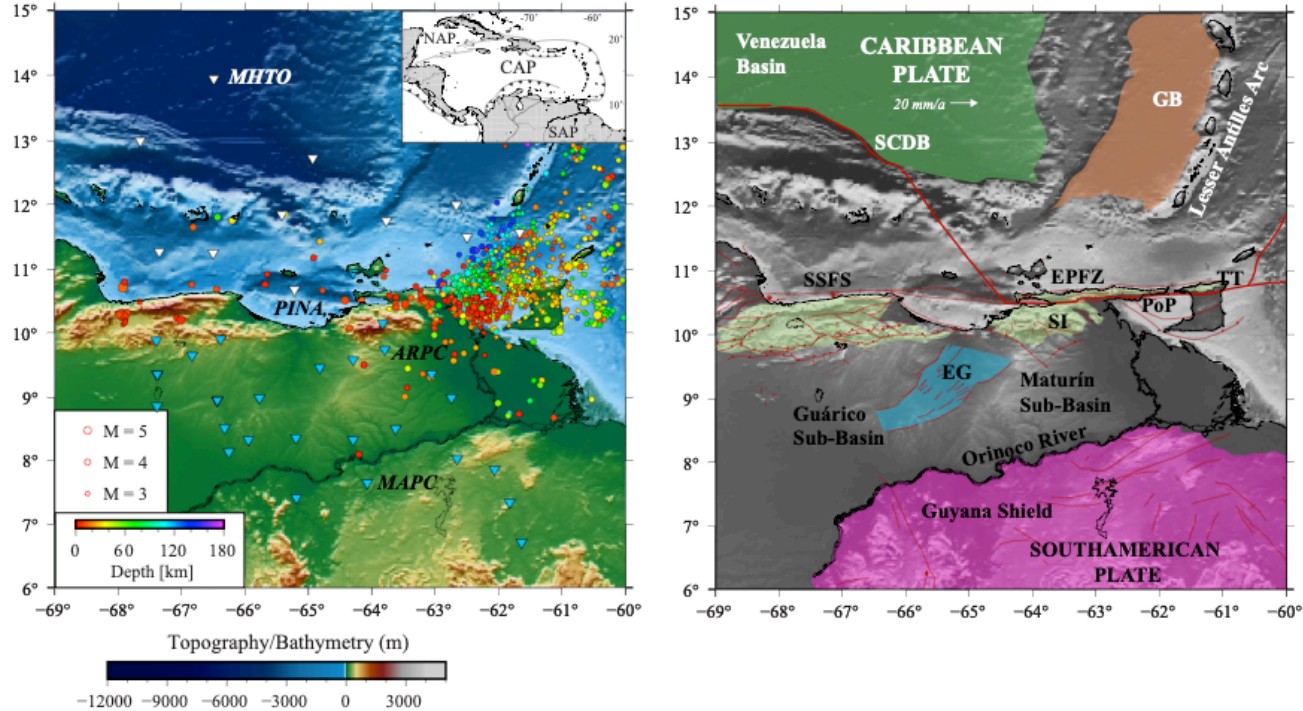

**Figure 1.** (Left) Topographic map of the studied area overlaid with stations (triangles) from the BOLIVAR project (2004 – 2005) and
seismicity (circles) from 2005-2020 ($M_w > 3.0$). On the top right is a view of the general context of the Caribbean Plate and Northern South
America. Blue triangles represent stations inland, while white triangles represent OBS. Labeled stations are later used to illustrate
methodology and results. (Right) Simplified geologic map of the studied region. The Precambrian shield is shown in purple, the Jurassic
Espino Graben (EG) in blue, in green the Serranía del Interior (SI), in dark green the Cretacic Venezuela Basin and in brown the Grenada
Basin. In red we show active faults as presented by Audemard et al (2006). Other abbreviations include: SSFS (San Sebastian Fault System),
EPFS (El Pilar Fault System), Trinidad and Tobago (TT) and Peninsula of Paria (PoP).





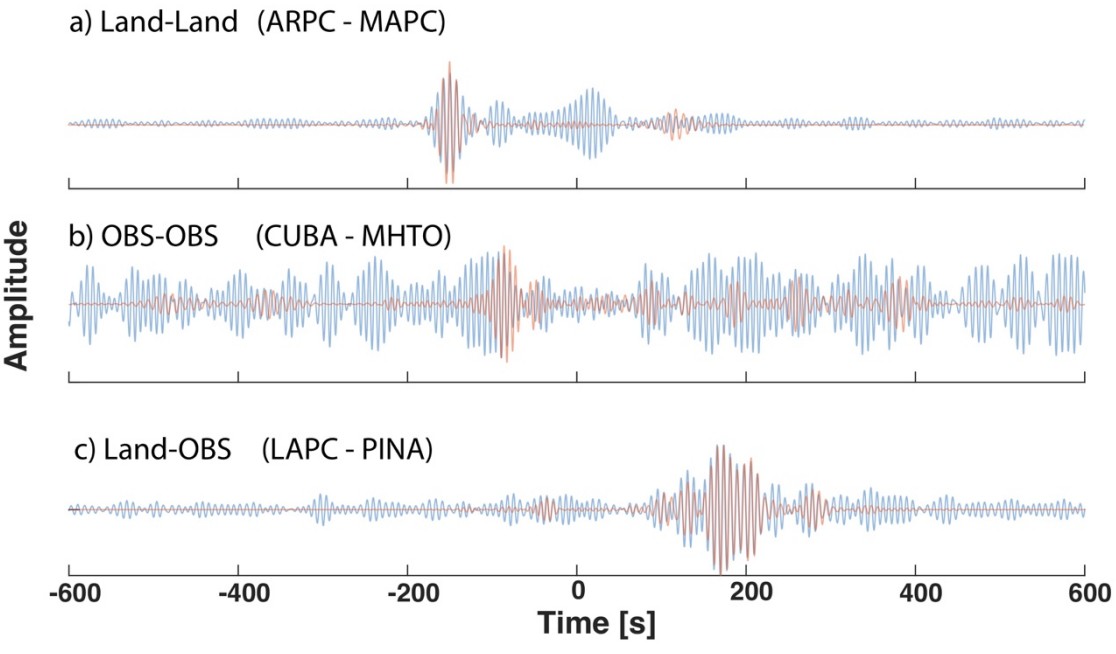

**Figure 2.** EGFs from three different cross-correlation pairs (EGFs) determined via linear stacking of cross-correlograms (blue) and ts-PWS stacking of the cross-correlograms (red). All EGFs are bandpass filtered over 6-10 s. This includes (a) EGFs from land stations ARPC to MAPC, (b) OBS stations CUBA to MHTO, and (c) LAPC to PINA, land to OBS stations, respectively. See Figure 1 for station locations.




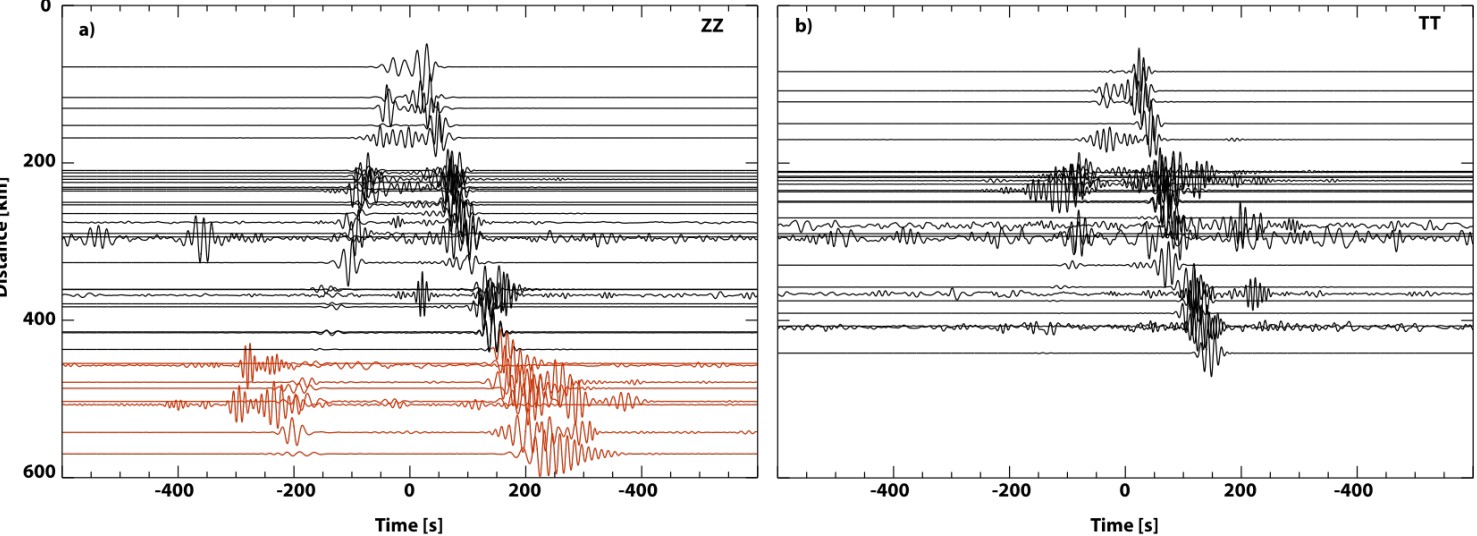



**Figure 3.** Record section of Guiana Shield station CMPC (see Figure 1). Specifically, we show the non-symmetric, stacked EGFs between
CMPC and all of the rest of stations. (a) Vertical-vertical (Z-Z) component where EGFs from CMPC to land stations are shown as black
lines and EGFs from CMPC to OBSs as red lines. (b) Same as (a), but for the Transverse-Transverse (T-T) component EGFs.



















**Figure 4.** Dispersion curves for multiple EGFs, station pair waveforms corresponding to those shown in Figure 2 (see Figure 1 for station locations). a) Rayleigh wave dispersion curves for phase velocity (dashed lines) and group velocity (solid lines). b) Love wave dispersion curves for phase velocity (dashed lines) and group velocity (solid lines). Note that only land-land stations are viable for Love wave dispersion analysis. c) and e) Phase velocity histograms of dispersion measurements for Rayleigh and Love waves, respectively. d) and f) Love velocity histograms of dispersion measures for Rayleigh and Love waves, respectively.




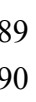


**Figure 5**. Rayleigh wave phase (upper panels, a-c) and group velocity (lower panels, d-f) maps for wave periods of T 12, 22 and 32 s. The gray box denotes the area with reliable measurements based on ray-path coverage.







**Figure 6.** Similar for Figure 5, but for Love wave phase and group velocity maps for wave periods 10, 15 and 20 s.






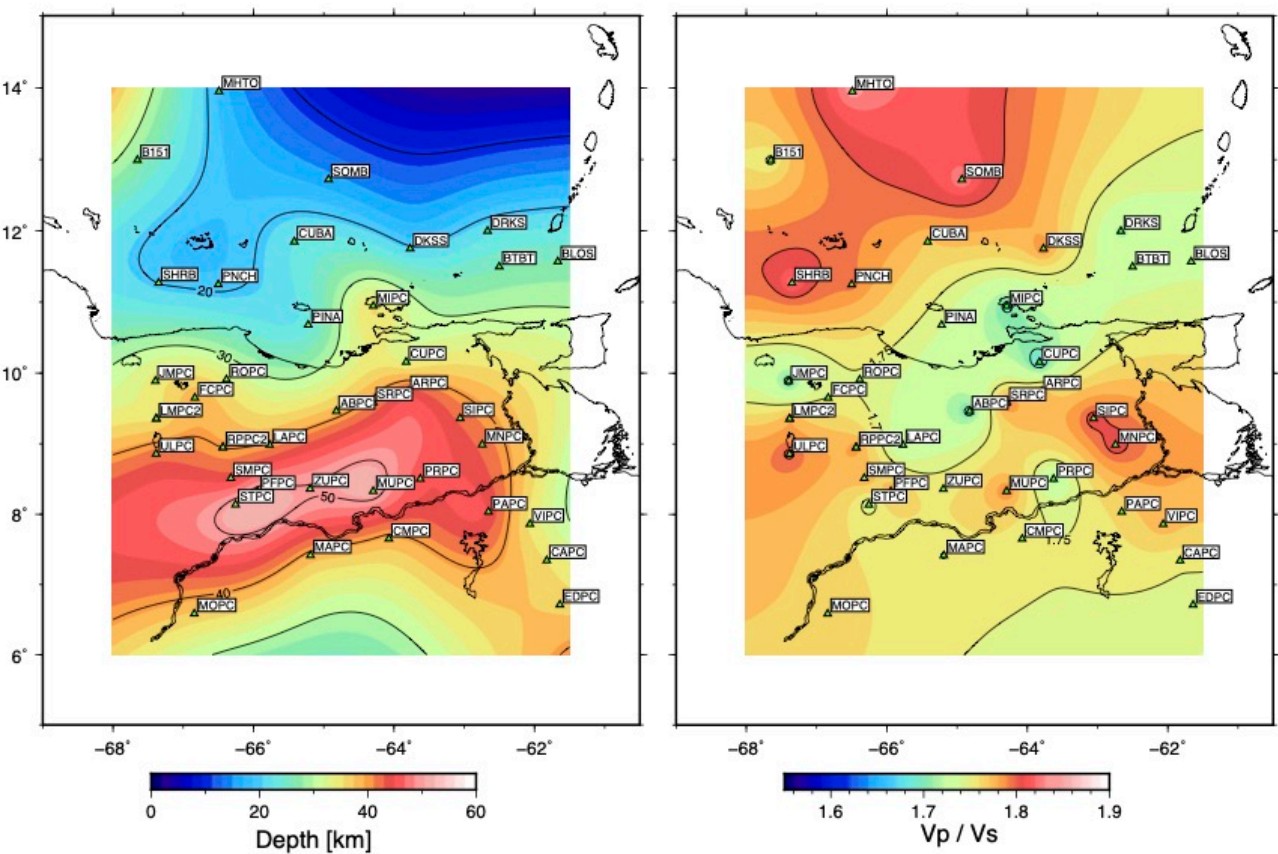

**Figure 7.** Map of the Moho depth estimation (left panel) and ratio Vp/Vs (right panel). Green triangles show the stations used to estimate
the Moho depth and the Vp / Vs ratio. Contour lines every 20 km for the Moho depth and in steps of 0.1 for the Vp/ Vs ratio. The inter-
station Moho depth have been retrieved thorough neighboring algorithm interpolation.

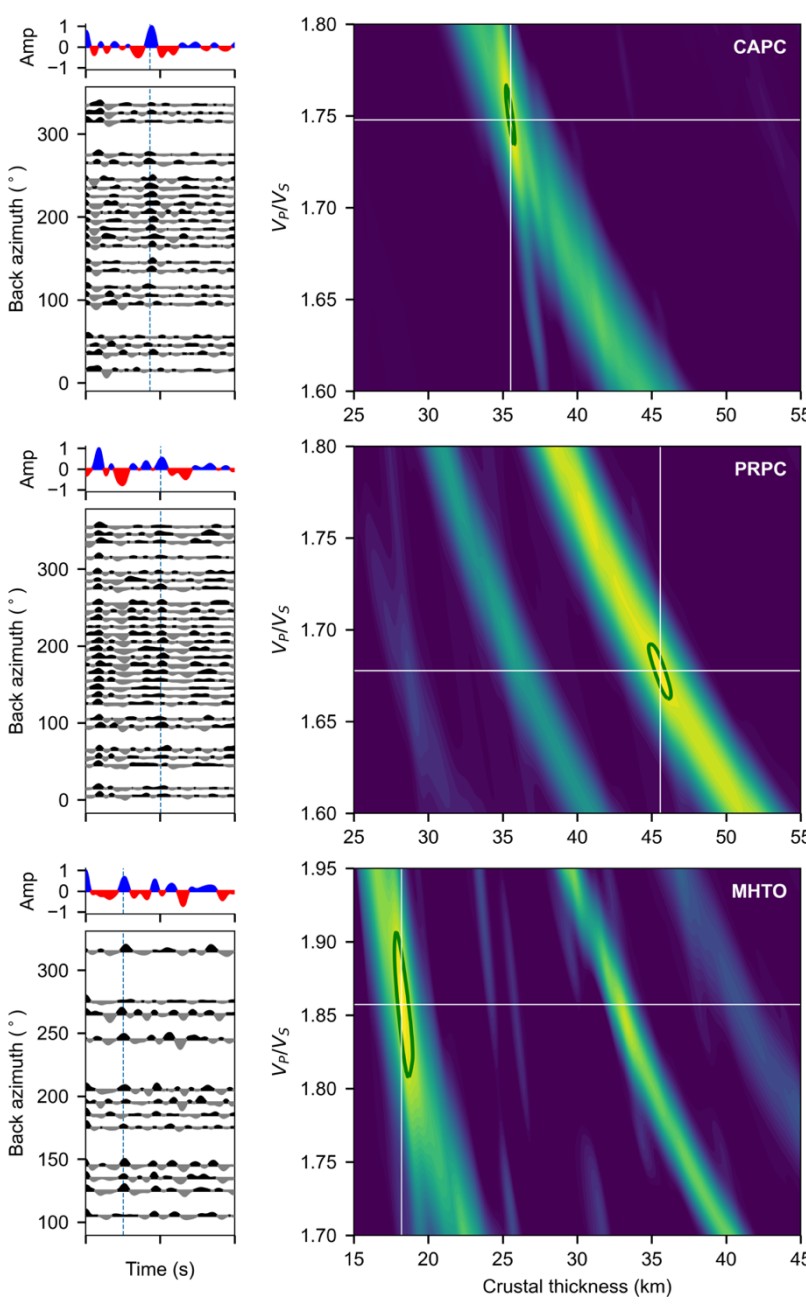

**Figure 8.** H-k and Receiver Function estimation for CAPC, STPC and PRCPC stations. Left panels: RFs of teleseismic earthquakes (black and white) used to obtain the RFs stack (blue and red). Right panels: H-k maps highlighting the maximum power corresponding to the optimal crustal thickness.

21
22
23

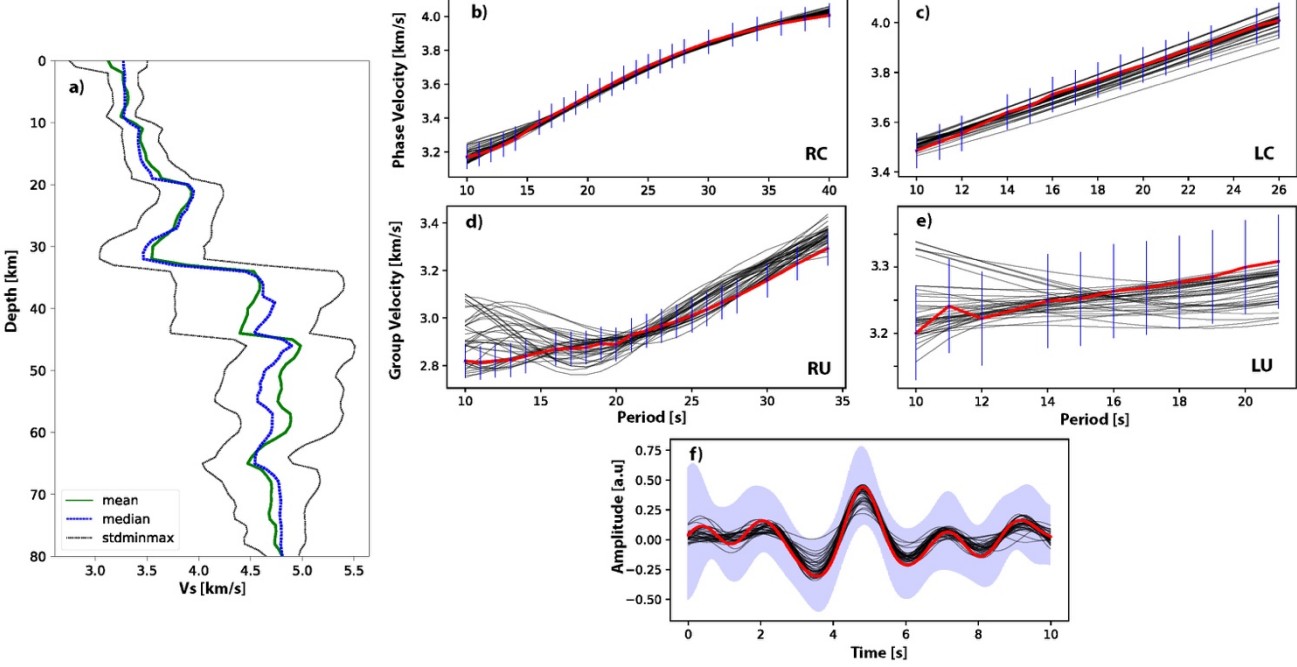

**Figure 9.** Illustrative example of 1D shear wave velocity inversion. a) green line represents the mean, the blue dots the median and the grey
lines the standard deviation estimated for 1D shear wave velocity model in (65.0° W -7.0 ° N, corresponding to the Guiana Shield) from the
ensemble of the best models per chain b) and d) phase and group velocity fit (grey line is the fit per chain) of the Rayleigh wavec), e) phase
and group velocity fit of the Love wave f) Receiver Function of station MAPC which was used to do the 1D shear wave inversion. Red line
the observed dispersion curves and receiver function, blue bars correspond to an error of three standard deviations and purple shadow is the
associated error of the receiver function.









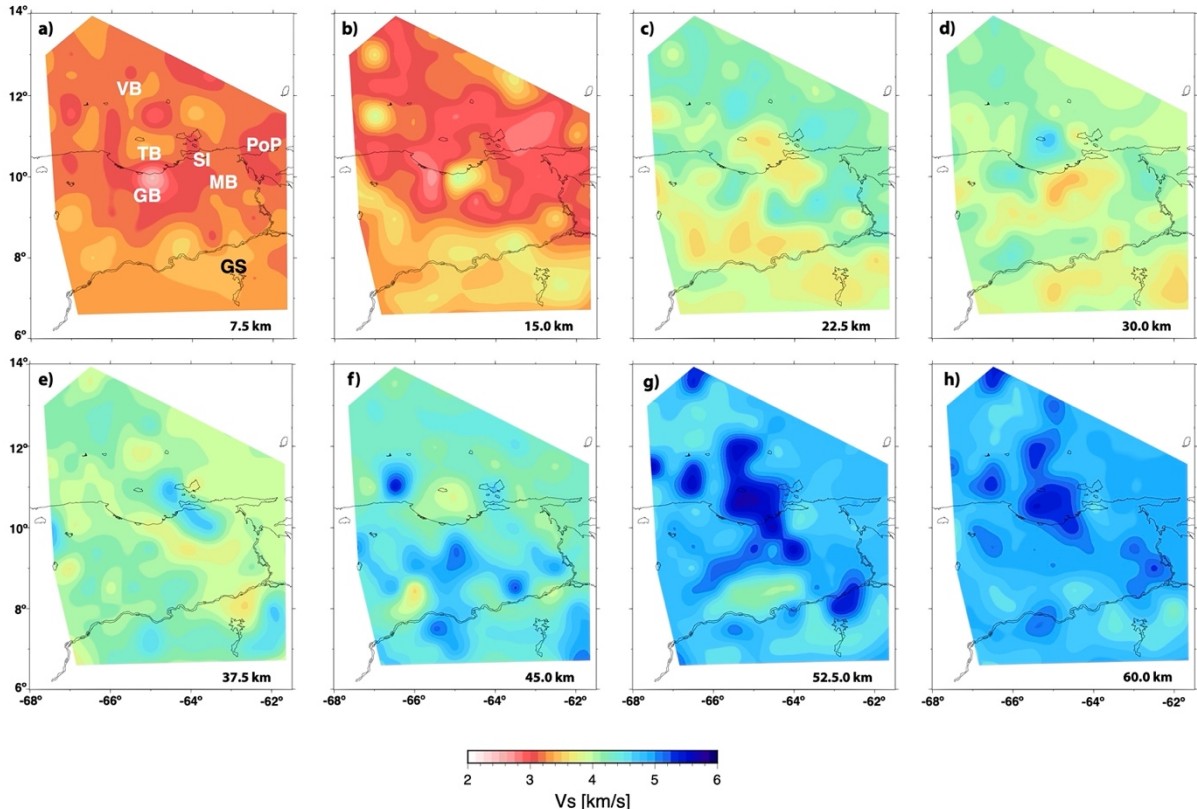


**Figure 10.** Horizontal slices of the 3D shear wave velocity model obtained in this study. From a) to h) increasing depth in steps of 7.5 km
until 60 km depth. VB (Venezuela Basin), GS (Guiana Shield), MB (Maturin Basin), GB (Guarico Basin), SI (Serrania del Interior), TB
(Turtle the Island-Barcelona Bay) and PoP (Peninsula of Paria).





**Figure 11.** Profiles show the Vs variations across the studied area. On the top left a map displaying the location of each profile. Each profile shows the Vs structure of the final model and the Moho depth (black dotted line) interpolated with the nearest neighbor algorithm. On the top, the topography and bathymetry from GEBCO 2021 (blue dashed line is 0 m.a.s.l.). Contours every 0.25 km/s. Abreviations stand for: BoB (Bonaire Basin), CdC (Cordillera de la Costa), GB (Guárico Basin), GS (Guiana Shield), MI (Margarita Island), SI (Serranía del Interior), MB (Maturín Basin), IT (Isla La Tortuga), CB (Cariaco Basin), EG (Espino Graben) and GreB (Grenada Basin).