# Peer review of "Upper lithospheric structure of northeastern Venezuela from joint inversion of surface wave dispersion and receiver functions"

_EGUsphere, 2022_

## Referee Comment (RC2)

[referee-annotated manuscript omitted]

---

## Author Comment (AC1)

Manuscript: **Upper lithospheric structure of northeastern Venezuela from joint inversion of surface wave dispersion and receiver functions**

We are grateful to RC1 to provide very constructive and thorough comments on our manuscript. We have responded in detail to each comment in the attached document "RC1_respond" in blue (RC1) text for each point individually and we have included the corrections related to the reviews (in blue) in a separated file "Manuscript corrections", corrected figures (Figs. 4, 5,6, 10 and 11) and new figures (Figs. S11 and S12 in the Supplementary Material) according to the reviews in the new Supplementary Material file.

**Reviewer 1**

This is a study in which the authors propose a new 3-D model of shear wave velocity (Vs) and moho depth in northern Venezuela using both receiver function on their own for Moho depth and a joint inversion of Rayleigh and Love phase and group velocity measurements obtained from noise cross-correlations, using both land-based and ocean-bottom seismometers. The authors use H-k stacking for measuring Moho depth and a linearised least-squares inversion to obtain surface wave dispersion curves and then use a hierarchical, transdimensional bayesian inversion scheme to jointly invert surface-wave data and receiver function data for shear wave velocity. The results show clear geographical coherence and known tectonic features. Overall, I think this is a good study that improves our knowledge of the area. However, I have some concerns about the methods and some figures need to be improved.

**Major Comments**

The authors use a joint inversion of receiver functions and surface-wave data, but they use receiver functions alone to measure the Moho depth. I do not understand why they perform a separate measurement for Moho depth instead of obtaining it from the joint inversion. Why use reciever functions in an inversion if not to better constrain the location of interfaces and especially the Moho? Figure 9 shows that the Moho is clearly visible on the inversion results. Consequently, the results of Moho depth measurements and the results of the Vs inversion as shown on Figure 11 do not seem to match, for example on profiles B-B' and D-D'. Also, some of the Moho depth measurements such as shown in Figure 8 for station PRPC are apparently not well constrained, maybe a joint inversion could have helped there.

**Answer:**

We agree with the reviewer that the crustal thickness obtained from the joint inversion is an interesting result. However, obtaining a crustal thickness map from the shear-wave velocity model is not straightforward, as the Moho velocities are not uniform across the model. For example, the shear wave velocity at the Moho in the Caribbean is ~4.0 km/s while it appears as ~4.25 km/s beneath the shield. This reflects the variation of mechanical properties of the lithospheric mantle (such as e.g., compositional variations) in the region. Hence, our preferred approach is to show the Vs model along with the Moho depths measured from the H-k stacking of the receiver functions alone to allow for a clearer comparison. Nevertheless, and in order to address the reviewer's comment, we have

included in the revised version of the manuscript a crustal thickness map picked from the shear-wave velocity inversion (Fig. S12 in the Supplementary Material). As explained in the main manuscript (see page 11, lines 17-21 of the revised manuscript), we find an overall good agreement between the two crustal thickness maps, only with some small disagreements. Indeed, it is expected that the results from both observations do not match exactly, as there is always some tradeoff in the joint inversion, with the different datasets "competing" for the better fit.

We would also like to emphasize that the availability of estimates of the crustal thickness from the H-k stacking (Zhu and Kanamori, 2000) of the receiver functions that agree with the previous geological knowledge allow us to impose further constraints upon the inversion (e.g., a broad min-max depth range for the Moho). These further constraints are helpful to discard geologically implausible models and perform a much more efficient search of the model space during the inversion procedure (e.g., Sambridge, 2001; Press, 1968). Moreover, we stress that the H-k stacking method of receiver functions is well-stablished, robust and is performed almost fully automatically, reducing the effects of possible analyst bias. To help clarify this issue, we have included an additional paragraph in section 2.5 (see page 8, lines 26-29):

*"To further reduce the non-uniqueness of the joint inversion problem and help to discard geologically implausible results, we restricted the Moho depths to a broad range (± 5 km) centered in the depths estimated from the H-k stacking of the receiver functions. These constraints on the Moho depths were applied to inversions performed at geographic points closer than 0.5° to the seismic stations."*

**Minor Comments**

Figure 1 is hard to read. The colorful background map and the many earthquakes make finding the stations challenging. As the authors are using seismic noise and teleseismic data, it is not clear why the local seismicity is shown. Figures 2 and 3 refer to Figure 1 for the location of stations CUBA, LAPC and CMPC, but those are not shown on the map. An indication of the location of the study area on the small map in the upper right corner would also be useful.

**Answer:** Agreed. We have added labels to the stations (MHTO, PINA, CUBA, ARPC, LAPC, MAPC and CMPC) that are referred to in subsequent figures and we also added a red square to the small map in the upper right corner clarifying the location of the study area. On the other hand, we find that it is important to show the local seismicity to illustrate the tectonic background and complexity of the study region. We added some text to the figure's caption to explain this purpose and this is also supported in the manuscript's introduction section:

Page 2, lines 25-28

*"The earthquakes in this cluster range from shallow to intermediate depths (~ 40 to 150 km), and their magnitudes vary from Mw 3 to 5, with a few relatively large events (Mw ≈ 6.5). The Paria cluster contains a gap in seismicity between 36-51 km depth that Clark et al. (2008) used to conclude that the subducting and buoyant pieces of the South American Plate occur along a near-vertical tear and support a "jelly sandwich" rheology."*

Page 4, line 99: The steps are listed in the wrong order, the authors first show how to retrieve Rayleigh and Love wave dispersion measurements, then how to obtain the RF and Moho depth measurements.

**Answer:** We agree that the overview text in section 2 did not clearly describe the processing steps in the same order as in the following sections. We have rewritten this part of the text as follows:

Page 4, lines 3-9

*"In the following sections, we briefly describe the steps that we followed to build the EGFs from the ambient seismic noise recordings. Overall, we have closely followed the approach described by Bensen et al. (2007) but with modifications in the preprocessing stage to obtain clear, reliable results for the OBSs. Then, we discuss how we retrieved measurements of group and phase velocities for both Rayleigh and Love waves from the EGFs, which we then used to build a set of phase and group velocity maps for the studied region. Next, the computation and analysis of the receiver functions is discussed. Finally, we describe the joint inversion of the information contained in the phase and group velocity maps and the receiver functions that we performed to obtain a 3D shear-wave velocity model of the studied area".*

Furthermore, we have also switched section 2.3 with section 2.4. In this way, we fully explain the extraction of the surface wave dispersion from the ambient noise recordings before moving on to the receiver functions and the joint inversion. The new order is:

2.2 Dispersion measurements of phase and group velocity

2.3 Phase and Group velocity maps

2.4 Station orientation and Moho Depth estimation

Page 5: It would be useful to indicate the direction of the main noise sources. This would make it easier to understand the kind of asymmetries and biases that are to be expected from the noise cross-correlations.

**Answer:**

While a detailed analysis of the noise sources is well beyond the scope of this study, we expect that the main noise generating sources might be from the Caribbean Sea, leading to non-symmetric EGFs. We have clarified this issue in the manuscript reviewed version:

Page 5, lines 7-11

*"Many of the noise cross-correlations in the study area (e.g., OBSs and land stations placed north Orinoco River) are non-symmetric. This is generally associated with the uneven distribution of ambient noise sources in the region (e.g. Webb 1998, Arnaiz-Rodríguez et al., 2021). In this case, we expect the*

*strongest pulses coming from the Caribbean Sea, where strong storms and hurricanes generally cross it in E-W direction"*

Figure 4: What are the red and blue bars on figures c-f? Why is the period scale logarithmic on figures a and b and linear on figures c-f? The caption needs to be clarified, like line 85: 'd) and f) Love velocity histograms of dispersion measures for Rayleigh and Love waves, respectively.'

**Answer:** Agreed. The caption of this figure has been modified to explain which is the difference between the blue and the red bars (additionally, we have incorporated the intermediate "clean" step). The period scale is linear in c-f because they are histograms that show statistics of the measures of the dispersion curves but a-b show examples of dispersion curves that conventionally are shown as a semi-log x axis to boost the visualization of short periods.

Page 6, line 60: 'The measurements of phase and group velocity from all station pairs at different periods', not 'in different periods'

**Answer:** Agreed. We thank the reviewer for this comment. We have corrected this grammatical mistake.

Figure S1: The caption is unclear: is this the output of the picking software?

**Answer:** Agreed. Fig S1 shows histograms and probability distribution of measures with respect to the velocity for Rayleigh, Love phase and group velocity, but there is no relation with a picking software. We have computed the Probability Distributions using NumPy library from Python 3.8 and plotted with Matplotlib. This section has been clarified in the Supplementary Material and we have added a caption in Fig S1.

Page 8, line 44: Did the authors use thinning for the McMC? 100 000 iterations on 40 cores does not seem like a lot for a joint transdimensional hierarchical bayesian inversion, would it be possible to show a density plot to show the inversion has fully converged?

**Answer:** Agreed. We have implemented a search over 40 chains and at each chain the algorithm makes an overall of 100.000 iterations (overall of 4000000 iterations per inversion point). We have clarified this issue in the following sentence from the main text:

Before:

*"The inversion was performed with 40 chains and each chain performed 100,000 iterations with a 2:1 for the burn-in and explorations phase respectively. Individual chains with a median likelihood that differs a threshold of 95% from the maximum likelihood of all chains were rejected."*

Page 9, lines 3-5
Now:

*"At each inversion point, we ran a total of 40 Markov chains with a final distribution of 100,000 iterations per chain, keeping all models within an accepting rate of 40% to form the posterior probability distribution, with a 2:1 for the burn-in and exploration phases, respectively."*

Furthermore, we have added two examples of probability density plots in the supplementary Material (Fig S11).

Page 9, line 84: Could the elongated feature seen on surface wave velocity maps be due to smearing? The resolution tests show some smearing in the area in the same direction.

**Answer:** We agree with the reviewer. Some of the elongated features found on the surface wave maps in the NE Orinoco Delta can be interpreted as smearing effects, likely due to the sparser ray coverage in the eastern end of the study area. For this reason our interpretation (i.e. Fig 11) focuses on the features west -63° where the raypath coverage and resolution are high and relatively well resolved. The sparser coverage in this region can be observed in the ray path figures in the Supplementary Material (Figures S2 and S3). The effects of this sparser coverage show both in our checkerboard tests (Figures S6 and S7) and in our spatial resolution maps (Figures S8 and S9). In particular, the latter always show the largest spatial resolution values in the southern and eastern limits of our study area. For this reason, our interpretation of the shear-wave velocity model (i.e. Figure 11) mainly focuses on the features west of 63°W where the ray path coverage is higher and the maps are relatively well-resolved. We have added a brief description of the resolution of the phase and group velocity maps at the end of section 3.1 to clarify this issue in the text (page 9, lines 24-29):

*"Overall, the spatial resolution values for the phase and group velocity maps remain stable at ~130-150 km in the central part of our studied area where the data coverage is high (Figs. S8 and S9 in the Supplementary Material). Some elongated features can be observed on the southeastern end of the studied area (around the Orinoco Delta), which could be interpreted as smearing effects due to the sparser data coverage in this region. Therefore, in the discussion section we will mainly focus on the features to the west of 63° where the phase and group velocity maps are well-resolved. "*

Figures 5 and 6: Would it be possible to show the relevant locations like in figure 10? Figure 1 is a long way and has a different size.

**Answer:** Agreed. We have added abbreviations of the relevant locations to Figures 5a and 6a to help with interpretation of the results and keep consistency with Figures 1 and Fig 10.

Page 14, line 16: shouldn't it be 'slab roll-back' rather than 'slab-roll back'?

**Answer:** We thank the reviewer for this observation and we have corrected this mistake.

Page 16, line 08: It would be necessary to cite the networks used in this study, the relevant information and DOI can be found on http://www.fdsn.org/networks/

**Answer:** Thank you for this review. We have found a reference for the dataset which we have incorporated to our reference list, including the DOI in the acknowledgements and citing the reference in  data section:

*"Frank Vernon, Gary Pavlis, Alan Levander, & Terry Wallace. (2003). Crust-Mantle Interactions during Continental Growth and High-Pressure Rock Exhumation at an Oblique Arc-Continent Collision Zone: SE Caribbean Margin [Data set]. International Federation of Digital Seismograph Networks. https://doi.org/10.7914/SN/XT_2003".*

---

## Author Comment (AC2)

Manuscript: **Upper lithospheric structure of northeastern Venezuela from joint inversion of surface wave dispersion and receiver functions**

We thank the reviewer for the time taken marking the manuscript and the figures. Corrections have been entered in the text "Manuscript_corrections" in green to make them easier to follow.

Reviewer 2:

The authors of this study propose a new 3D model of shear wave velocity (Vs) and moho depth of eastern Venezuela, from the Caribbean Basin in the North to the Guiana Shield in the South, using both reciever function alone for Moho depth imaging and a joint inversion of Rayleigh and Love phase and group velocity measurements obtained from noise cross- correlations, using an amphibious (land-ocean) seismometer array. This 3D model is build from 1D profiles spaced 0.5°x0.5°. The authors use H-k stacking for measuring Moho depth and a linearised least-squares inversion to obtain surface wave dispersion curves and then use a hierarchical, transdimensional bayesian inversion scheme to jointly invert surface-wave data and reciever function data for shear wave velocity. The results show clear geographical coherence and known geologic features. Overall, this study seems to improve knowledge of the area. However,even if my field of expertise is not the use of earthquake waves to image Earth Interior, I have three major concerns about this contribution:

1) Some figures need to be improved. Particularly, Figure 1 (left panel) should only show seismicity used during this evaluation. Minor comments on the other figures are provided in an annotated pdf.

**Answer:** We have updated the figures accordingly with the reviewer's comments see new Figs. 1, 4, 5, 6, 10 and 11 in the revised manuscript) and new figures (Figs. S11 and S12 in the Supplementary Material). We have checked and updated the bibliography accordingly and have included more references in the discussion (see pages 12, 13, 14, 15 and 16 of the revised manuscript). This has strengthened the interpretation of our Vs model. Furthermore, we have made sure to add the active faults and Precambrian tectonic provinces in most figures to ease their interpretation. We must make clear that the events in Fig 1 are just for explaining the tectonic and seismic of the region, they are not used in this study, therefore events for a long range of dates are shown (we added some explanatory text to the figure's caption).

Page 2, lines 25-28

"The earthquakes in this cluster range from shallow to intermediate depths (~ 40 to 150 km), and their magnitudes vary from Mw 3 to 5, with a few relatively large events (Mw ≈ 6.5). The Paria cluster contains a gap in seismicity between 36-51 km depth that Clark et al. (2008) used to conclude that the subducting and buoyant pieces of the South American Plate occur along a near-vertical tear and support a "jelly sandwich" rheology."

2) Some previous internationally-reviewed studies on crustal structure of the same region by local researchers (e.g., Schmitz et al, 2005, 2021), using other methods (e.g. wide- angle data) are curiously not cited. Of course, comparison of results between those different studies (with different approaches: active seismics) is not presented.

**Answer:** We have updated our discussion to include these references and have compared our results to theirs. We would like to point out that the two main previous works (Niu et al., 2007 and Schmitz et al., 2021) show important discrepancies and our results lie in the middle between those two studies. We also note that new data is required in Eastern Venezuela to further understand the crustal architecture of that region.

3) the authors keep the comparison of their results to other similar (geophysical/seismological: passive seismics) studies: Niu et al.; Miller et al.; Masy et al.; Arnaiz et al., and so on. It would seem that they are well aware of the Bolivar Project results. However, even the aim of the paper being the correlation/imaging/identification of geological/tectonic/geodynamic features, little referencing of geological studies is applied. For instance, the Espino Graben geometry and its Cenozoic-Quaternary southward directed inversion is well known from oil-industry seismics and other studies.

**Answer:** We have included several new references throughout the manuscript and especially in the Discussion section to address these shortcomings. The new key references and additions to the original text are highlighted in green in the updated manuscript.

Minor comments, typo and form corrections are provided in annotated pdf of the contribution.

**Answer:** We have reviewed the entire manuscript, figures and captions to correct all the mistakes pointed out by the reviewer.